# The Involvement of l-Arginine-Nitric Oxide-cGMP-ATP-Sensitive K^+^ Channel Pathway in Antinociception of BBHC, a Novel Diarylpentanoid Analogue, in Mice Model

**DOI:** 10.3390/molecules26247431

**Published:** 2021-12-08

**Authors:** Hui Ming Ong, Ahmad Farhan Ahmad Azmi, Sze Wei Leong, Faridah Abas, Enoch Kumar Perimal, Ahmad Akira Omar Farouk, Daud Ahmad Israf, Mohd Roslan Sulaiman

**Affiliations:** 1Department of Biomedical Sciences, Faculty of Medicine and Health Sciences, Universiti Putra Malaysia, Serdang 43400, Selangor, Malaysia; oguiming85@gmail.com (H.M.O.); farhanazmirahman@gmail.com (A.F.A.A.); enoch@upm.edu.my (E.K.P.); ahmadakira@upm.edu.my (A.A.O.F.); daudaia@upm.edu.my (D.A.I.); 2UPM-MAKNA Cancer Research Laboratory, Institute of Bioscience, Universiti Putra Malaysia, Serdang 43400, Selangor, Malaysia; frederick_Leong@hotmail.com; 3Department of Food Sciences, Faculty of Food Science & Technology, Universiti Putra Malaysia, Serdang 43400, Selangor, Malaysia; faridah_abas@upm.edu.my; 4Natural Medicines and Product Research Laboratory, Institute of Bioscience, Universiti Putra Malaysia, Serdang 43400, Selangor, Malaysia

**Keywords:** antinociceptive, BBHC, cGMP, diarylpentanoid, nitric oxide, ATP-sensitive potassium channel

## Abstract

The present study focuses on the possible involvement of l-arginine-nitric oxide-cGMP-ATP-sensitive K^+^ channel pathway in the antinociceptive activity of a novel diarylpentanoid analogue, 2-benzoyl-6-(3-bromo-4-hydroxybenzylidene)cyclohexen-1-ol (BBHC) via a chemical nociceptive model in mice. The antinociceptive action of BBHC (1 mg/kg, i.p.) was attenuated by the intraperitoneal pre-treatment of l-arginine (a nitric oxide synthase precursor) and glibenclamide (an ATP-sensitive K^+^ channel blocker) in acetic acid-induced abdominal constriction tests. Interestingly, BBHC’s antinociception was significantly enhanced by the i.p. pre-treatment of 1H-[1,2,4]oxadiazolo[4,3-a]quinoxalin-1-one (ODQ), a selective inhibitor of soluble guanylyl cyclase (*p* < 0.05). Altogether, these findings suggest that the systemic administration of BBHC is able to establish a significant antinociceptive effect in a mice model of chemically induced pain. BBHC’s antinociception is shown to be mediated by the involvement of l-arginine-nitric oxide-cGMP-ATP-sensitive K^+^ channel pathway, without any potential sedative or muscle relaxant concerns.

## 1. Introduction

Nitric oxide (NO) is a membrane permeable, free radical gas that has been identified as a potent biological mediator with diverse physiologic functions including synaptic transmission and central sensitization in the nociceptive process [1,2]. Nitric oxide and its byproduct called l-citrulline are synthesized from l-arginine (a nitric oxide precursor) through an enzymatic reaction by neuronal or non-neuronal nitric oxide synthases (NOS) [3,4,5]. Tissue injury or inflammation activates peripheral afferent nociceptive neurons and propagates the impulses to the dorsal horn of the spinal cord. The resulting excitability in the spinal area is called a central sensitization. During central sensitization, N-methyl-D-aspartate (NMDA) receptors are activated to amplify the release of glutamate from presynaptic endings in the spinal cord and further increases the intracellular level of calcium ions. This action also causes the stimulation of calmodulin-sensitive nitric oxide synthase [6,7,8].

The induction of nitric oxide synthase results in elevated NO production, which reacts directly with downstream targets of the neurons or diffuse out from the neurons to affect neighboring tissues [1]. The production of NO activates soluble guanylyl cyclase (sGC), a major receptor for nitric oxide. This activation of sGC causes the conversion of guanosine triphosphate (GTP) to cyclic guanosine monophosphate (cGMP) [2]. It then modulates its downstream cellular targets such as cGMP-dependent protein kinase, receptors and ion channels [6]. Previous research shows that the syntheses of NO and cGMP are important in the adenosine triphosphate (ATP)-sensitive K^+^ channel activation [9,10]. The opening of ATP-sensitive K^+^ channels via the l-arginine-NO-cGMP pathway allows the negative modulation of neuronal excitability and results in an efflux of potassium ions, thus leading to antinociception [11,12,13].

Diarylpentanoids are bioactive compounds with a 5-carbon spacer developed from a synthetic modification of curcumin. Diarylpentanoids have exhibited superior anti-inflammatory properties by suppressing tumor necrosis factor alpha (TNF-α), interleukins, nitric oxide, prostaglandins, lipoxygenases and cyclooxygenases [14,15,16,17,18]. They also show excellent antioxidant properties protecting cells from 2,2-diphenyl-1-picrylhydrazyl (DPPH) free radicals, superoxide and hydroxyl radicals [15]. Diarylpentanoids have resolved the issues of poor stability and bioavailability seen with curcumin [19]. Diarylpentaoids exhibit a great chemical stability at a physiological pH and metabolic stability in rat liver microsomes, making them the most promising candidates among other curcuminoids to develop novel bioactive compounds as an alternative for curcumin with enhanced stability and bioavailability [20,21].

Extensive focus has been given to diarylpentanoids from the scientific community for their superb pharmacological properties and enhanced bioavailability in comparison to curcumin [22]. We had previously demonstrated that a newly synthesized compound named 2-benzoyl-6-(3-bromo-4-hydroxybenzylidene)cyclohexen-1-ol (BBHC), from a novel family of diarylpentanoids possessed significant inhibitory activity on NO production in interferon-γ/lipopolysaccharide-induced RAW 264.7 macrophages. The in vitro study of BBHC against NO production had strongly suggested its potential as an NO inhibitor with an IC_50_ value of 15.2 µM [23]. However, no in vivo investigation has ever been performed to confirm the suppressive activity of BBHC towards NO synthesis. Therefore, the present study was conducted to verify the possible involvement of l-arginine-NO-cGMP-ATP-sensitive K^+^ channel pathway in the antinociceptive activity caused by BBHC.

## 2. Results

### 2.1. Involvement of l-Arginine-Nitric Oxide

As shown by the results in Figure 1, the pre-treatment of l-arginine (a nitric oxide precursor, 100 mg/kg, i.p.) significantly antagonized the antinociceptive effects of L-NOARG (a nitric oxide synthase inhibitor, 20 mg/kg, i.p.) and BBHC (1 mg/kg, i.p.) in acetic acid-induced abdominal constriction test with *p* < 0.001 and *p* < 0.01 respectively.

### 2.2. Involvement of Cyclic Guanosine Monophosphate

The result presented in Figure 2 shows that the pre-treatment of ODQ, a selective inhibitor of soluble guanylyl cyclase (2 mg/kg, i.p.) significantly enhanced the antinociception of BBHC (1 mg/kg, i.p.) in the acetic acid-induced abdominal constriction test with *p* < 0.05 when comparing to BBHC-only treated group.

### 2.3. Involvement of ATP-Sensitive K^+^ Channel

Figure 3 shows the pre-treatment of glibenclamide (10 mg/kg, i.p., an ATP-sensitive K^+^ channel inhibitor) significantly reversed the antinociceptive activity of BBHC (1 mg/kg, i.p.) in the acetic acid-induced abdominal constriction model with *p* < 0.01 as compared to BBHC-only treated group.

### 2.4. Rota-Rod Test

In Figure 4, the intraperitoneal treatment of BBHC (1.0 mg/kg) did not modify the motor coordination of the mice as compared to the control group (vehicle, i.p.) The performance time was significantly reduced in the diazepam-treated mice (4 mg/kg, i.p.) group as compared with the control animals in a rota-rod test.

## 3. Discussion

In this study, we aimed to characterize the possible mechanisms of action employed by BBHC to elicit its antinociceptive activity. From the result we obtained from the present study, it was shown that the pre-administration of l-arginine (a substrate for nitric oxide synthase or also known as a nitric oxide precursor), at a dose that did not cause any significant pain by acetic acid injection, significantly reversed the antinociceptive activities of BBHC and L-NOARG (a nitric oxide synthase inhibitor). Together with the findings from the previous study [23], we could confirm that the antinociception produced by BBHC is very likely to involve the l-arginine-nitric oxide pathway.

Nitric oxide is a diffusible gas which is permeable to the cellular membrane; therefore, it is generally involved in the nociceptive pathway at peripheral, dorsal horn and cerebral cortex levels [24,25]. l-arginine is an amino acid which serves as a precursor in nitric oxide synthesis [26]. Several previous studies have implicated that the activation of nitric oxide production by the systemic administration of l-arginine results in hyperalgesia and the release of inflammatory mediators such as bradykinin and substance P, which in turn enhances the nociceptive and inflammatory experiences [27,28].

There are three isoforms of nitric oxide synthase, namely neuronal NOS (nNOS), endothelial (eNOS) and inducible NOS (iNOS) [6]. In the present experiment, a nitric oxide synthase inhibitor, called l-NOARG (N^ω^-nitro-l-arginine) was used to generate attenuation in nitric oxide production. l-NOARG is an active inhibitor for nNOS and eNOS (calcium-dependant NOS), which is commonly used to interrupt the synthesis of constitutive nitric oxide [10,29,30]. Nitric oxide generated by nNOS is highly upregulated in the neurones located in dorsal horn of the spinal cord upon stimulation by noxious stimuli at the periphery [31,32]. On the other hand, nitric oxide catalyzed by eNOS is found to play a role in the modulating acute inflammation [33]. Administration of BBHC and l-NOARG alone exhibited significant antinociceptive activities against acetic acid-induced pain responses but the pre-treatment of l-arginine antagonized the antinociception produced by BBHC and l-NOARG significantly. This result shows that BBHC and l-NOARG may share some common routes of inhibition towards nociceptive pain as well as further strengthening the involvement of nitric oxide in the BBHC-induced antinociceptive mechanism.

The rapid production of nitric oxide activates its receptors, known as sGC, and consequently elevates the level of cGMP, which in turn modulates various physiological functions such as nociception and antinociception [34]. However, several studies have also demonstrated the stimulation of the NO-cGMP pathway which causes hyperalgesia rather than analgesia, whereby in such cases, the systemic introduction of a nitric oxide synthase inhibitor resulted in antinociceptive activities in a dose-dependent manner [35,36,37,38]. The involvement of the NO-cGMP pathway in nociceptive or inflammatory effects has been shown upon its activation by substance P, bradykinin and carrageenin [39].

The NO-cGMP pathway activation is dependent on the production of NO by NOS, which subsequently induces NO-sensitive sGC to form cGMP, the most crucial mediator of the pathway [40]. Previous findings have suggested that the transducing power of nitric oxide in signal transmission may be mediated by cGMP upon the activation of guanylyl cyclase. Hence, it may suggest an important role of cGMP in mediating the upregulation or downregulation of nociceptors in cases of nociceptive pathways [34]. The co-administration of an sGC inhibitor with glutamate was found to antagonize the glutamate-induced hyperalgesia at the spinal level, indicating that the hyperalgesia effect of glutamate is extensively mediated by the release of NO and the activation of soluble guanylyl cyclase [41].

ODQ is an inhibitor of nitric oxide-activated soluble guanylate cyclase, which is commonly used to prevent the catalytic action of soluble guanylyl cyclase in the conversion of cGMP from guanosine triphosphate [41]. This is agreed by our results that the intraperitoneal injection of ODQ significantly showed an antinociceptive response in acetic acid-induced pain. More interestingly, the co-administration of an analgesic dose of ODQ with BBHC had significantly enhanced the BBHC’s antinociceptive activity against the algesic condition induced by acetic acid in the animals. The enhancement of BBHC-induced antinociception may implicate the combined inhibitory effects of BBHC and ODQ against nitric oxide synthase and soluble guanylyl cyclase activities respectively. This may explain the improvement in analgesia by the co-administration of BBHC and ODQ in the present test. Therefore, the antinociceptive activity of BBHC has implied the involvement of the NO-cGMP pathway in acetic acid-induced nociception.

A large amount of studies has mentioned the close interaction between the NO-cGMP pathway and the ATP-sensitive K^+^ channel, whereby the activation of nitric oxide and cGMP can stimulate the opening of ATP-sensitive K^+^ channels [42,43,44,45,46]. It is evident that the opening of ATP-sensitive K^+^ channels can produce repolarization or even hyperpolarization of the cell membrane, leading to the reduction of membrane excitability [42,47]. Our present results have shown that the administration of glibenclamide, an ATP-sensitive K^+^ channel inhibitor, significantly antagonized the antinociceptive effect of BBHC in acetic acid-induced nociception model. Glibenclamide has been reported to block ATP-sensitive K^+^ channel in particular without apparent actions on other types of potassium channels like voltage-gated and calcium ion-dependent K^+^ channels [42,48,49,50]. Therefore, these findings provide elucidation that BBHC displays its analgesic property via the opening of ATP-sensitive K^+^ channels, which in turn permits potassium ion efflux from the neurone cells and causes the repolarization of cell membranes. This cascade of events establishes a situation of reduced excitability in the membrane.

The systemic administration of BBHC (1 mg/kg, i.p.) failed to cause any motor dysfunction in the rota-rod. Thus, the possible non-specific muscle relaxant properties and sedative effects in BBHC-induced antinociceptive activities, which are eliminated in the present study. On the whole, these findings suggest that the antinociceptive activities produced by BBHC are mediated by its inhibition on the NO-cGMP pathway and ATP-sensitive K^+^ channel-opening action.

## 4. Materials and Methods

### 4.1. Synthesis of BBHC

The compound, BBHC, was synthesized by Dr. Leong Sze Wei in the Laboratory of Natural Products, Institute of Bioscience, Universiti Putra Malaysia, Serdang, Malaysia.

#### 4.1.1. Synthetic Scheme of BBHC

As shown in synthetic scheme above (Figure 5), BBHC was prepared by a series of reactions [23], which involved benzoylation of cyclohexanone and aldol condensation of aromatic aldehyde. The benzoylation of cyclohexanone was carried out through Stork enamine acylation with a Dean-Stark distillation set-up. Accordingly, cyclohexanone was first reacted with pyrrolidine to afford *N*-(1-cyclohexenyl)pyrrolidine (I), an essential enamine intermediate for benzoylation. The synthesized intermediate I was then reacted with benzoic anhydride to afford 2-benzoylcyclohexanone (II) crude product. The 2-benzoylcyclohexanone was purified by column chromatography and further reacted with 3-bromo-4-hydroxybenzaldehyde in acidic aldol condensation conditions to prepare the desired compound.

#### 4.1.2. General Procedure for Synthesis of I and II

Based on the previously described protocol [23], a catalytic amount of *p*-toluenesulphonic acid was added into a mixture of cyclohexanone (20 mmol) and pyrrolidine (20 mmol) in 30 mL of toluene kept in 100 mL single neck round bottom (SNRB) flask at room temperature. The mixture was then refluxed on a Dean & Stark trap for 2 h to prepare I. Upon completion, 20 mmol of benzoic anhydride in 20 mL of toluene was added dropwise into the reaction solution (I) and stirred for 24 h. Distilled water (10 mL) was then added and further refluxed for 30 min. The resulting reaction mixture was extracted thrice with 3 M HCl and once with 20 mL water. The toluene layer was dried over an anhydrous magnesium sulphate and concentrated in vacuo to yield II. The crude was purified through column chromatography.

#### 4.1.3. General Procedure for the Synthesis BBHC

Then, the synthesis of BBHC was continued as described previously [23]. In a 50 mL SNRB flask, 2-benzoylcyclohexanone (5 mmol) and 3-bromo-4-hydroxybenzaldehyde (5 mmol) were dissolved in 30 mL of acetic acid. Catalytic amount of concentrated sulfuric acid was added and the reaction mixture was stirred overnight. The resulting mixture was extracted with ethyl acetate and washed with 10% sodium bicarbonate solution. The organic layer was then dried over anhydrous magnesium sulphate and evaporated using rotary evaporator (Rotavapor^®^R-300System, Heidolph Instruments GmbH & CO. KG, Schwabach, Germany). The targeted compound was purified by column chromatography (Merck silica gel 60, mesh 70–230 and elution with 95% hexane: 5% ethyl acetate) to produce BBHC. The structure (Figure 6) and the purity of the compound were identified and characterized by using ^1^H-NMR and ^13^C-NMR (Varian 500 MHz, Varian Inc., Palo Alto, CA, USA), HPLC utilizing Waters Xbridge C18 column (5 µm, 150 mm × 4.6 mm) (Thermo Finnigan Surveyor, San Josè, CA, USA) and gas chromatography mass spectrometry (Shimadzu GCMS-QP5050A Mass Spectrometer, Shimadzu, Kyoto, Japan). The purity of the compound was 98.85%.

### 4.2. Preparation of BBHC

BBHC was kept chilled (±16 °C) in the Physiology Laboratory, Faculty of Medicine and Health Sciences, Universiti Putra Malaysia. A total of 1 mg/kg of BBHC was freshly prepared and dissolved in a vehicle solution (5% DMSO, 5% Tween 20 and 90% normal saline) for each test.

### 4.3. Preparation of Drugs and Chemicals

l-arginine, N^ω^-nitro-l-arginine (l-NOARG), 1H-[1,2,4]oxadiazolo[4,3-a]quinoxalin-1-one (ODQ), glibenclamide and acetic acid were purchased from Sigma Chemical Co. (St Louis, MO, USA). Acetic acid, l-arginine, l-NOARG, glibenclamide and diazepam were dissolved in normal saline (0.9% NaCl). ODQ was dissolved in 5% DMSO.

### 4.4. Experimental Animals

Experiments were carried out on male ICR mice (20–30 g, 3–4 weeks old with an average age of 3.83 weeks ± 0.39) throughout the study. The male ICR mice were purchased from the Agrovet Resources Sdn. Bhd. (845599-V), Malaysia and were habituated in the Animal House of Faculty of Medicine and Health Sciences, Universiti Putra Malaysia at room temperature with 12 h of light/dark cycle with free access to food pellets and water ad libitum. The mice were subjected to acclimatization to the laboratory condition for at least 7 days before starting a test. Every animal was only used once throughout this study. All the experimental work reported in this study was carried out in fulfillment of the current guidelines for the care of laboratory animals, including the ethical guidelines for experimental pain investigations in conscious animals, with approval from the Institutional Animal Care and Use Committee, Universiti Putra Malaysia (UPM/IACUC/AUP-R066/2015). Data were collected in a blinded, randomized, and controlled design for all the tests in this study. For every animal, an investigator would administer the treatment based on a randomization table (using online QuickCalcs software) and another investigator would observe the animal behaviors without knowing the treatment given to the animal.

### 4.5. Involvement of l-Arginine-Nitric Oxide

The determination of possible involvement of l-arginine-nitric oxide in BBHC’s antinociceptive activity was evaluated according to the method as previously described but with slight amendments [25]. The mice were randomly divided into 6 groups with 6 mice in each group (*N* = 36). The 6 groups of mice (*n* = 6) were classified as: vehicle, l-arginine, l-NOARG, l-arginine + l-NOARG, BBHC and l-arginine + BBHC. The mice were pre-treated with l-arginine (100 mg/kg, i.p., a nitric oxide precursor) 15 min before the injections of BBHC (1 mg/kg, i.p.) or l-NOARG (20 mg/kg, i.p., a nitric oxide synthase inhibitor). The control group only received vehicle (10 mL/kg, i.p.). After 30 min, the mice were injected with 0.8% of acetic acid (10 mL/kg, i.p.) for the initiation of pain. Animals were immediately placed in an individual observation chamber after acetic acid injection. The counting of abdominal constriction behavior began 5 min after acetic acid injection and this continued for 30 min.

### 4.6. Involvement of Cyclic Guanosine Monophosphate

The investigation of a potential contribution of cyclic guanosine monophosphate (cGMP) in BBHC-related analgesia was carried out with minor modifications [40,51]. The mice were randomly divided into 4 groups with 6 mice in each group (*N* = 24). The 4 groups of mice (*n* = 6) were classified as: vehicle, ODQ, BBHC and ODQ + BBHC. Prior to pain induction, the mice were pre-treated with ODQ (2 mg/kg, i.p., a selective inhibitor of soluble guanylyl cyclase) 15 min before receiving BBHC (1 mg/kg, i.p.) or vehicle (10 mL/kg, i.p.). The control group only received vehicle (10 mL/kg, i.p.). After 30 min, the mice were observed for their analgesic responses in acetic acid-induced (0.8%, 10 mL/kg, i.p.) abdominal constriction test as described previously.

### 4.7. Involvement of an ATP-Sensitive K^+^ Channel

The possible participation of ATP-sensitive K^+^ channel in the antinociceptive property of BBHC was investigated with minor changes [52]. The mice were randomly divided into 4 groups with 6 mice in each group (*N* = 24). The 4 groups of mice (*n* = 6) were classified as: vehicle, glibenclamide, BBHC and glibenclamide + BBHC. The mice were pre-treated with glibenclamide (10 mg/kg, i.p., an ATP-sensitive K^+^ channel inhibitor) 15 min before the mice received BBHC (1 mg/kg, i.p.) or vehicle (10 mL/kg, i.p.). The control group only received a vehicle (10 mL/kg, i.p.). The mice were injected with 0.8% acetic acid (10 mL/kg, i.p.) 30 min post-treatment with BBHC or a vehicle.

### 4.8. Rota-Rod Test

To find out the possible non-specific sedative properties of BBHC, a rota-rod test was carried out [53] with minor modifications. The mice were randomly divided into 3 groups with 6 mice in each group *(N* = 18). The 3 groups of mice (*n* = 6) were classified as: vehicle, BBHC and diazepam. The animals were exposed to the practice on a rota-rod apparatus (Ugo Basile, model 47600) at 20 rpm one day before the rota-rod test. On the actual day of the test, the mice were placed on the rota-rod apparatus (20 rpm), 30 min after the intraperitoneal pre-treatments with vehicle (10 mL/kg), BBHC (1.0 mg/kg) or diazepam (4 mg/kg). The mice on the rota-rod apparatus were observed and recorded for their performance latency over a period of 120 s.

### 4.9. Statistical Analysis

The data obtained were expressed as means ± S.E.M of 6 mice. The differences were analyzed by One-way ANOVA followed by the Tukey’s multiple comparison test as post hoc test via GraphPad-Prism 5.0 software (GraphPad Software, San Diego, CA, USA). Differences between means were considered statistically significant when *p* < 0.05.

## 5. Conclusions

In the present study, BBHC inhibits nitric oxide production significantly, indicating an indirect deactivation of NOS. In addition, BBHC is speculated to be involved in the cGMP pathway to enhance analgesia. BBHC also shows the potential to activate ATP-sensitive K^+^ channels, which subsequently reduces membrane excitability. In conclusion, these results suggest that BBHC exerts an antinociceptive activity via the modulation of l-arginine-NO-cGMP-ATP-sensitive K^+^ channel pathway. Further research on the other possible mechanisms of action of BBHC’s antinociceptive activity is currently underway in our laboratory.

## Figures and Tables

**Figure 1 molecules-26-07431-f001:**
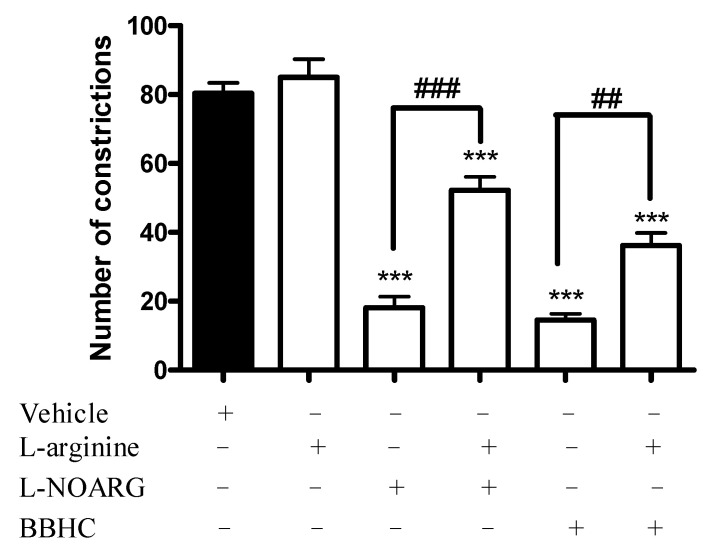
The involvement of l-arginine-nitric oxide in BBHC-induced antinociceptive activity in acetic acid-induced abdominal constriction test. Each column represents the mean ± S.E.M. of 6 mice. The mice were pre-treated with l-arginine (nitric oxide precursor, 100 mg/kg, i.p.) followed by administration with BBHC (1 mg/kg, i.p.) or L-NOARG (nitric oxide synthase inhibitor, 20 mg/kg, i.p.) before acetic acid injection. The control group only received vehicle (10 mL/kg, i.p.). The asterisks denote the significance levels as compared with control, *** *p* < 0.001; the hashes denote the significance level as compared with BBHC-only or L-NOARG-only treated group, ^##^ *p* < 0.01, ^###^ *p* < 0.001, by one-way ANOVA followed by Tukey’s post hoc test.

**Figure 2 molecules-26-07431-f002:**
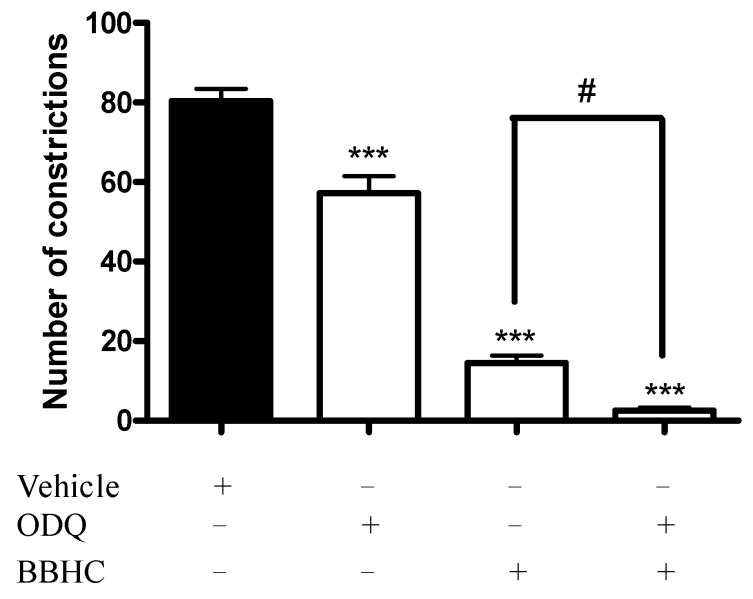
The involvement of cGMP in BBHC-induced antinociceptive activity in acetic acid-induced abdominal constriction test. Each column represents the mean ± S.E.M. of 6 mice. The mice were pre-treated with ODQ (selective inhibitor of soluble guanylyl cyclase, 2 mg/kg, i.p.) followed by administration with BBHC (1 mg/kg, i.p.) before acetic acid injection. The control group only received vehicle (10 mL/kg, i.p.) The asterisks denote the significance levels as compared with control, *** *p* < 0.001; the hashes denote the significance level as compared with BBHC-only treated group, ^#^ *p* < 0.05, by one-way ANOVA followed by Tukey’s post hoc test.

**Figure 3 molecules-26-07431-f003:**
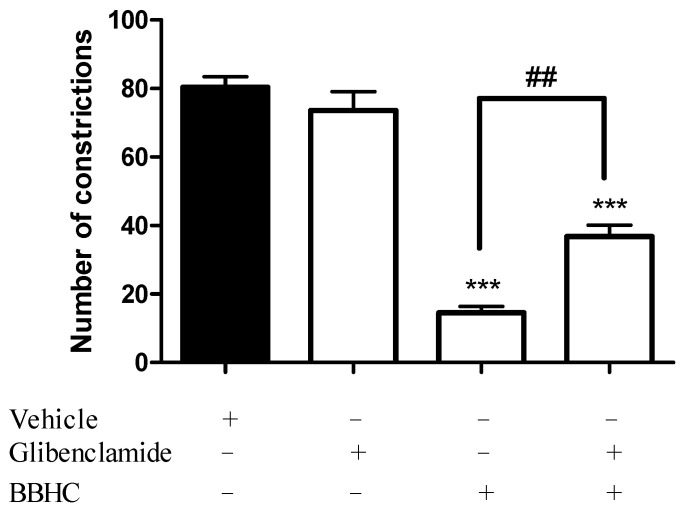
The involvement of ATP-sensitive K^+^ channel in BBHC-induced antinociceptive activity in acetic acid-induced abdominal constriction test. Each column represents the mean ± S.E.M. of 6 mice. The mice were pre-treated with glibenclamide (ATP-sensitive K^+^ channel inhibitor, 10 mg/kg, i.p.) followed by administration with BBHC (1 mg/kg, i.p.) before acetic acid injection. The control group only received vehicle (10 mL/kg, i.p.) The asterisks denote the significance levels as compared with the control, *** *p* < 0.001; the hashes denote the significance level as compared with the BBHC-only treated group, ^##^ *p* < 0.01, by one-way ANOVA followed by Tukey’s post hoc test.

**Figure 4 molecules-26-07431-f004:**
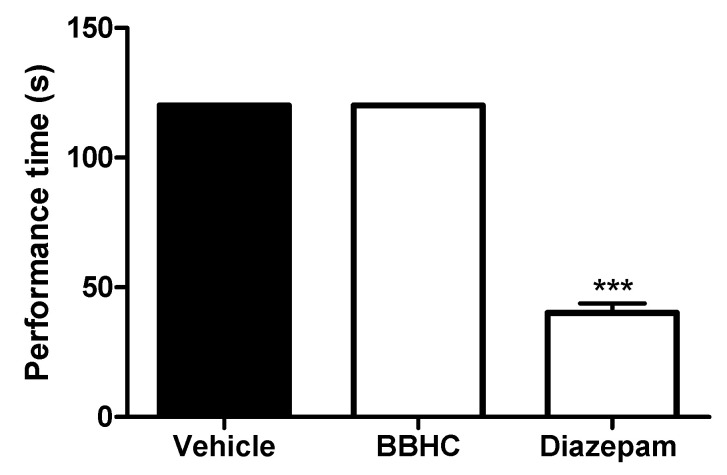
The effect of BBHC in the rota-rod test in mice. Each column represents the mean ± S.E.M. of 6 mice. The mice were pre-treated with control (vehicle, 10 mL/kg, i.p.), BBHC (1.0 mg/kg, i.p.) or diazepam (4 mg/kg, i.p.) before being subjected to a rota-rod test. The asterisks denote the significance levels as compared with control, *** *p* < 0.001 by One-way ANOVA followed by Tukey’s post hoc test.

**Figure 5 molecules-26-07431-f005:**
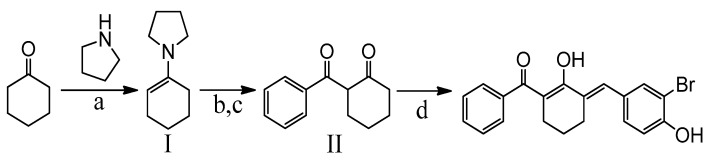
Reagents and conditions: (**a**) *p*-toluene-sulphonic acid, toluene, reflux (2 h); (**b**) benzoic anhydride, room temperature (24 h); (**c**) H_2_O, reflux (0.5 h); (**d**) 3-bromo-4-hydroxybenzaldehyde, acetic acid, H_2_SO_4_, RT (overnight).

**Figure 6 molecules-26-07431-f006:**
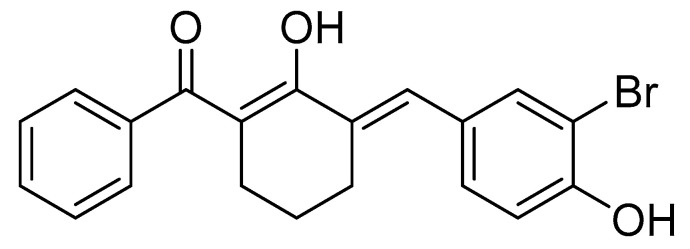
The chemical structure of BBHC. BBHC: 2-benzoyl-6-(3-bromo-4-hydroxybenzylidene)cyclohexen-1-ol. [Colour: Yellow; Yield: 65.19%; m.p.: 172–173 °C; Mass calculated: 384.0361; Mass found: 384.0379 ^1^H NMR (500 MHz, chloroform-d) δ ppm 1.69 (quin, *J* = 6.04 Hz, 2 H) 2.50–2.57 (m, 2 H) 2.71–2.78 (m, 2 H) 5.70 (br. s., 1 H) 7.05 (d, *J* = 8.45 Hz, 1 H) 7.34 (dd, *J* = 8.45, 2.04 Hz, 1 H) 7.41–7.50 (m, 3 H) 7.55–7.61 (m, 3 H) 7.63 (s, 1 H) 16.74 (s, 1 H).^13^C NMR (126 MHz, chloroform-d) δ ppm 23.5, 27.1, 27.6, 108.3, 110.3, 116.0, 127.6, 128.1, 130.5, 130.6, 131.4, 131.5, 131.9, 133.6, 138.2, 152.2, 176.2, 195.2].

## Data Availability

The data presented in this study are available on request from the corresponding author.

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
