# Peer review of "The Involvement of l-Arginine-Nitric Oxide-cGMP-ATP-Sensitive K+ Channel Pathway in Antinociception of BBHC, a Novel Diarylpentanoid Analogue, in Mice Model"

_molecules, 2021, doi:10.3390/molecules26247431_

Round 1

Reviewer 1 Report

This is a well-done manuscript dealing with a novel and interesting topic of the underlying mechanism of the antinociceptive activity of one of the curcumin derivatives. The study adds important information to the profile of BBHC that could greatly help to get more benefit as a therapeutic agent in several health disorders. Nevertheless, some concerns need to be addressed to fit for publication as follows:
1.    Title: the tested animal species should be clarified. Thus, it is highly recommended to add "in mice model".
2.    The use of abbreviations throughout the manuscript should be revised. E.g. line 22 the full term of ODQ should be first mentioned then the abbreviation should be used further. The same issue has been repeated for NMDA in line38 and many times throughout the manuscript.
3.    Introduction:
-    Line 56-62: very long paragraph.
-    Line 62-64: rephrase for clarity.
-    Line 70: the authors stated, "We had previously" By revising Ref 14, the paper not belongs to the authors of the manuscript. 
-    Ref 14 was mentioned in line 73 then again in line 75. One mention is enough.
-    The authors should give more information on BBHC or diarylpentaoids as this is the main core of the study. In addition, the exact bioavailability of BBHC compared to curcumin should be mentioned.
4.    Discussion needs to be refined by avoiding repetition of the information already mentioned in the abstract and focus on the interpretations of the findings of the study.
5.    Material and methods:
-    Lines 74 and 76: CHLOROFORM. Revise.
-    Line 289: the age of mice should be presented as means±SE. also, clarify the age of mice used.
-    In each experiment, clarify the total number of mice used and group classification.
6.     A major concern in the current manuscript is the updating of the references instead of the recent studies on the topic. The authors should update the references by studies in 2019-2021.

Author Response

Response to Reviewer 1 Comments

Point 1: Title: the tested animal species should be clarified. Thus, it is highly recommended to add "in mice model".

Response 1: The phrase ‘in mice model’ is included in the new title as recommended by Reviewer 1.

Point 2: The use of abbreviations throughout the manuscript should be revised. E.g. line 22 the full term of ODQ should be first mentioned then the abbreviation should be used further. The same issue has been repeated for NMDA in line38 and many times throughout the manuscript.

Response 2: The use of abbreviations throughout the manuscript was revised as advised.

Point 3: Introduction: Line 56-62: very long paragraph.

Response 3: Line 56-62 are shortened as advised (line 56-60 in revised manuscript).

Point 4: Introduction: Line 62-64: rephrase for clarity.

Response 4: Line 62-64 are rephrased as advised (line 60-61 in revised manuscript).

Point 5:  Introduction: Line 70: the authors stated, "We had previously" By revising Ref 14, the paper not belongs to the authors of the manuscript.

Response 5: We apologize for the mistake for citing the wrong reference and correction has been made. The correct reference for lines 68-73 (in revised manuscript) is no. 23 in the reference list.

Point 6: Introduction: Ref 14 was mentioned in line 73 then again in line 75. One mention is enough. 

Response 6: Reference 14 in line 71 is removed as advised (in revised manuscript). As mentioned in response 5, the reference for lines 68-73 (in revised manuscript) is actually Ref. no. 23 in the reference list.

Point 7:
Introduction: The authors should give more information on BBHC or diarylpentaoids as this is the main core of the study. In addition, the exact bioavailability of BBHC compared to curcumin should be mentioned.

Response 7: Thanks for your suggestion. The exact bioavailability of BBHC compared to curcumin was not done in the present study. However, diarylpentanoids generally exhibited better stability at physiological pH and metabolic stability in rat liver microsomes. Therefore, BBHC is predicted to have enhanced stability and bioavailability than curcumin.

Point 8: Discussion needs to be refined by avoiding repetition of the information already mentioned in the abstract and focus on the interpretations of the findings of the study.

Response 8: Discussion is revised by removing the repetitive information that was already mentioned in introduction.

Point 9: Material and methods: Lines 274 and 276: CHLOROFORM. Revise.

Response 9: The word ‘CHLOROFORM’ is revised to ‘chloroform’ in line 276 and 278 as advised (line 266 and 268 in revised manuscript).

Point 10: Material and methods: Line 289: the age of mice should be presented as means±SE. also, clarify the age of mice used.

Response 10: The age of mice with mean ± SE is stated in line 281-282 as advised (in revised manuscript).

Point 11: Material and methods: In each experiment, clarify the total number of mice used and group classification.

Response 11: The total number of mice used and group classification for each experiment are stated in line 300-302, 312-314, 322-324 and 331-333 (in revised manuscript).

Point 12: A major concern in the current manuscript is the updating of the references instead of the recent studies on the topic. The authors should update the references by studies in 2019-2021.

Response 12: Thanks for your suggestion. All the papers that we cited are relevant and applicable to our study, especially the references we used in the materials and methods section are the optimized procedures that published by our colleagues in the same research group. We sincerely hope to maintain all the references in the manuscript.

Reviewer 2 Report

Some suggestions are given below to make this manuscript readable and clearer to audience.  

Abstract

Line 25: Could you please rephrase this sentence? “that could be mediated by involvement…”. This sentence is to long, could you please split up that sentence to help the reader to better understand the idea.

Introduction:

Line 33: Could you please change “sensitization” to “sensitization”, here and in other parts of the manuscript

Line 36: Could you please use “Resulting from” instead of “Resulted from”

Line 37: This sentence can be rephrased to make it clear to readers       

Line 38: Could you please give the meaning of the abbreviation “NMDA” receptors the first time that it appears in the manuscript.

Lines 42 and 43: In this phrase, the word “Nitric Oxide” is used thrice, so it is redundant, could you please try to rephrase this sentence to avoid this issue.

Line 44 and 45: I find the idea in this sentence unclear. Are the authors trying to say “The production of NO, activates guanylyl cyclase” or “The production of NO, triggers guanylyl cyclase activation”

Line 45 to 47: Could you please make this sentence clear? This is because the first part of the sentence is redundant.

Line 47: Please, interchange the noun “cGMP” by pronoun, avoiding redundancy

Lines 49 to 50: This sentence needs to improve syntaxis and grammar

Line 51: Could you please use “Allows” instead “permits”

Line 54: Syntaxis mistake, please correct this sentence. Suggestion: “Diarylpentanoids are bioactive compounds with a 5-carbon spacer…”

Lines 57: Syntaxis mistake, when authors conjugate present perfect, they use two similar verbs: “shown” and “to exhibit”. Please, eliminate one… I suggest: “have exhibited”

Line 59: take off the space before “, lipoxygenase”… and please, add an “s” to make it a plural

Line 62: This sentence is not necessary “Not only do diarylpentanoids possess the same beneficial properties of curcumin”

Figure 5, Line 231: RT… make it explicit…

Line 166: “L-NOARG” typing mistake

Line 238: add a comma after the phrase “The synthesized intermediate I”

Line 239 and 240: If “2-benzoylcyclohexanone” was purified by column chromatography, make it explicit.

Line 246: Please, spell out the abbreviation “SNRB” the first time it is used.

Line 247: Use “trap” instead the word “apparatus”

Line 252: use “yield” instead the word “give”

Line 260: word “rotatory” need to be changed by “rotary”

Line 262: Add “y” to “Column chromatography”

Line 293:  “z” instead “s” in the word  “acclimatization”

Line 295: “work” instead “works”

Line 308: use the phrase “underwent treatment with” instead “subjected to”

Line 310: Use the Word “started” or “began” instead “commenced”

Line 333: In this sentence, authors need to make clear that groups (i.e. vehicle, BBHC and diazepam mice) were composed by 6 mice each

Line 344 and 345: When authors make this statement “addition, BBHC prevented the catalytic action of soluble guanylyl cyclase and interrupted the cGMP pathway” I think that is a little bit risky. This is because it appears to refer about inhibition of soluble guanylyl cyclase, and authors didn’t provide references about in-vitro inhibition of this enzyme by BBHC. Yes, they demonstrate an additive in-vivo antinociceptive effect, but that doesn’t necessarily involve soluble guanylyl cyclase catalysis inhibition. On the other hand, in lines 197 – 201 (Discussion), authors propose about NO-cGMP pathway involvement, which in turn is broader possibility. They need to make it clearer because this can be confusing.

Could you please define how mice randomization was carried out in general?. Could you please review some references used in this work (mainly at materials and methods section), where some authors that make reference to other authors. Is it possible to cite the original work and authors that developed these methods?.

Could you please rephrase the conclusion section to make it clearer?

Author Response

Response to Reviewer 2 Comments

Point 1: Abstract: Line 25: Could you please rephrase this sentence? “that could be mediated by involvement…”. This sentence is too long, could you please split up that sentence to help the reader to better understand the idea.

Response 1: The sentence in line 25 is rephrased as in line 25-27 of the revised manuscript.

Point 2: Introduction: Line 33: Could you please change “sensitisation” to “sensitization”, here and in other parts of the manuscript

Response 2: The word ‘sensitisation’ is changed to ‘sensitization’ in the revised manuscript as advised.

Point 3: Introduction: Line 36: Could you please use “Resulting from” instead of “Resulted from”

Response 3: The phrase ‘resulted from’ is changed to ‘resulting from’ in the revised manuscript as advised.

Point 4: Introduction: Line 37: This sentence can be rephrased to make it clear to readers      

Response 4: The sentence in line 37 is rephrased as suggested (line 37-39 in revised manuscript).

Point 5: Introduction: Line 38: Could you please give the meaning of the abbreviation “NMDA” receptors the first time that it appears in the manuscript.

Response 5: The meaning of the abbreviation ‘NMDA’ is stated in line 38 for its first appearance in the manuscript.

Point 6: Introduction: Lines 42 and 43: In this phrase, the word “Nitric Oxide” is used thrice, so it is redundant, could you please try to rephrase this sentence to avoid this issue.

Response 6: The sentence in line 42 and 43 is rephrased as advised (line 44-46 in revised manuscript). However, we have to keep the first and second ‘nitric oxide’ in the sentence as the first ‘nitric oxide’ is referring to ‘nitric oxide synthase’.

Point 7: Introduction: Line 44 and 45: I find the idea in this sentence unclear. Are the authors trying to say “The production of NO, activates guanylyl cyclase” or “The production of NO, triggers guanylyl cyclase activation”

Response 7: We meant to say ‘the production of NO activates soluble guanylyl cyclase’. This phrase is rephrased in the revised manuscript (line 46-47 in revised manuscript).

Point 8: Introduction: Line 45 to 47: Could you please make this sentence clear? This is because the first part of the sentence is redundant.

Response 8: The sentence in line 45-47 is rephrased as suggested (line 47-48 in revised manuscript).

Point 9: Introduction: Line 47: Please, interchange the noun “cGMP” by pronoun, avoiding redundancy

Response 9: The term ‘cGMP’ in line 47 is changed to pronoun ‘it’ (line 48 in revised manuscript).

Point 10: Introduction: Lines 49 to 50: This sentence needs to improve syntaxis and grammar

Response 10: The sentence is phrased with improved syntaxis and grammar (line 50-51in revised manuscript).

Point 11: Introduction: Line 51: Could you please use “Allows” instead “permits”

Response 11: The word ‘permits’ in line 51 is changed to ‘allows’ (line 52 in revised manuscript).

Point 12: Introduction: Line 54: Syntaxis mistake, please correct this sentence. Suggestion: “Diarylpentanoids are bioactive compounds with a 5-carbon spacer…”

Response 12: The sentence in line 54 is rephrased as suggested (line 55 in revised manuscript).

Point 13: Introduction: Lines 57: Syntaxis mistake, when authors conjugate present perfect, they use two similar verbs: “shown” and “to exhibit”. Please, eliminate one… I suggest: “have exhibited”

Response 13: The sentence in line 57 is rephrased as suggested (line 57-58 in revised manuscript).

Point 14: Introduction: Line 59: take off the space before “, lipoxygenase”… and please, add an “s” to make it a plural

Response 14: The sentence in line 59 is changed as suggested (line 58 in revised manuscript).

Point 15: Introduction: Line 62: This sentence is not necessary “Not only do diarylpentanoids possess the same beneficial properties of curcumin”

Response 15: The sentence in line 62 is edited as advised (line 61-62 in revised manuscript).

Point 16: Figure 5, Line 231: RT… make it explicit…

Response 16: The abbreviation ‘RT’ is replaced with ‘room temperature’ for Figure 5.

Point 17: Line 166: “L-NOARG” typing mistake

Response 17: The typing mistake in line 166 is corrected (line 160 in revised manuscript).

Point 18: Line 238: add a comma after the phrase “The synthesized intermediate I”

Response 18: The comma is added as advised (line 230 in revised manuscript).

Point 19: Line 239 and 240: If “2-benzoylcyclohexanone” was purified by column chromatography, make it explicit.

Response 19: The sentence in line 239-240 is rephrased. We meant to say ‘2-benzoylcyclohexanone’ was purified by column chromatography (line 231-232 in revised manuscript).

Point 20: Line 246: Please, spell out the abbreviation “SNRB” the first time it is used.

Response 20: The abbreviation ‘SNRB’ in line 246 is spelled out for the first time it is used (line 238 in revised manuscript).

Point 21: Line 247: Use “trap” instead the word “apparatus”

Response 21: The word ‘apparatus’ in line 247 is changed to ‘trap’ (line 239 in revised manuscript).

Point 22: Line 252: use “yield” instead the word “give”

Response 22: The word ‘give’ in line 252 is changed to ‘yield’ (line 244 in revised manuscript).

Point 23: Line 260: word “rotatory” need to be changed by “rotary”

Response 23: The word ‘rotatory’ in line 260 is changed to ‘rotary’ (line 253 in revised manuscript).

Point 24: Line 262: Add “y” to “Column chromatography”

Response 24: The alphabet ‘y’ in line 262 is added to ‘column chromatograph’ (line 254 in revised manuscript).

Point 25: Line 293:  “z” instead “s” in the word  “acclimatization”

Response 25: The word ‘acclimatisation’ in line 293 is changed to ‘acclimatization’ (line 286 in revised manuscript).

Point 26: Line 295: “work” instead “works”

Response 26: The word ‘works’ in line 295 is changed to ‘work’ (line 288 in revised manuscript).

Point 27: Line 308: use the phrase “underwent treatment with” instead “subjected to”

Response 27: The phrase ‘subjected to’ in line 308 is changed to ‘injected with’ (line 306 in revised manuscript) because the acetic acid was used to induce pain.

Point 28: Line 310: Use the Word “started” or “began” instead “commenced”

Response 28: The word ‘commenced’ in line 310 is changed to ‘began’ (line 308 in revised manuscript).

Point 29: Line 333: In this sentence, authors need to make clear that groups (i.e. vehicle, BBHC and diazepam mice) were composed by 6 mice each

Response 29: The number of mice in each treatment group is clarified for line 333 (line 331-333 in revised manuscript).

Point 30: Line 344 and 345: When authors make this statement “addition, BBHC prevented the catalytic action of soluble guanylyl cyclase and interrupted the cGMP pathway” I think that is a little bit risky. This is because it appears to refer about inhibition of soluble guanylyl cyclase, and authors didn’t provide references about in-vitro inhibition of this enzyme by BBHC. Yes, they demonstrate an additive in-vivo antinociceptive effect, but that doesn’t necessarily involve soluble guanylyl cyclase catalysis inhibition. On the other hand, in lines 197 – 201 (Discussion), authors propose about NO-cGMP pathway involvement, which in turn is broader possibility. They need to make it clearer because this can be confusing.

Response 30: We agree with the comment from the reviewer. Thus, we have changed to conclude the involvement of NO-cGMP in a more general way as advised (line 346-347 in revised manuscript).

Point 31: Could you please define how mice randomization was carried out in general? Could you please review some references used in this work (mainly at materials and methods section), where some authors that make reference to other authors. Is it possible to cite the original work and authors that developed these methods?.

Response 31: Randomization is defined as shown in line 293-296 (in revised manuscript). The references we used in the materials and methods section are the optimized procedures that published by our colleagues in the same research group. We would hope to maintain these references for materials and methods.

Point 32: Could you please rephrase the conclusion section to make it clearer?

Response 32: The conclusion is rephrased as advised (line 345-352 in revised manuscript).

Reviewer 3 Report

In this manuscript, the authors explored the effects of 2-benzoyl-6-(3-bromo-4-hydroxybenzylidene)cyclohexen-1-ol (BBHC) on chemical nociceptive model in mice. They show that BBHC-induced antinociceptive activity was mimicked by L-NOARG, nitric oxide synthase inhibitor and was canceled by L-arginine (Figure 1) or glibenclamide, an ATP-sensitive K+ channel inhibitor (Figure 3) and was enhanced by ODQ, a selective inhibitor of soluble guanylyl cyclase (Figure 2). They also show that BBHC itself do not appear sedative properties (Figure 4). Based on the above results, they propose that antinociception produced by BBHC is very likely to INHIBIT the L-arginine-NO-sGC-ATP-sensitive K+ channels pathway.

The authors previously reported that Zerumbone, a bioactive sesquiterpene induced antinociceptive effect, being reversed by L-arginine and mimicked by L-NOARG, nitric oxide synthase inhibitor (10). However, other studies have indicated that activation of the NO–cGMP– ATP-sensitive K+ channels pathway causes anti-nociceptive effects in neuropathic pain [9, 11-13]. This discrepancy may be still not elucidated.

Again, it was shown that 1) the peripheral L-arginine–nitric oxide–cyclic GMP pathway and ATP-sensitive K+ channels and opioid receptors are involved in the antinociceptive effect (9) and 2) the opening of K+ channels in the peripheral and central nervous system are important mechanisms in the antinociception induced by many types of drugs and natural products (11) and3) PDE-5 inhibitor enhances the antinociceptive effect of morphine (12) and 4) the activation of the K+ channel enhances the antinociceptive effects of fentanyl (13). The present data show that the NO–cGMP- ATP-sensitive K+ channels pathway has pronociceptive effect which is controversial as described above.

I have some concerns to address as below,

  1. The authors state that we had previously demonstrated that a newly synthesised compound named, 2-benzoyl-6-(3-bromo-4-hydroxybenzylidene)cyclohexen-1-ol (BBHC), from a novel family of diarylpentanoids possessed significant inhibitory activity on NO production in IFN-γ/LPS-induced RAW 264.7 macrophages [14] (lanes 70-73). But the authors do not show up and misread the contents of this paper.
  2. They focused on the L-arginine-nitric oxide-cGMP-ATP-sensitive K+ channel pathway. Thus, they should determine the effect of BBHC on NOSs enzyme activity in vitro.
  3. Is the antinociceptive effect produced by BBHC reversed by naloxone, an opioid receptor antagonist?
  4. They would be better off using 7-Nitroindazole asides from L-NOARG to test the involvement of neuronal NOS on the antinociceptive effect.
  5. What is the difference mechanisms and potency between BBHC and Zerumbone (10)?
  6. How they interpretate the discrepancy of the NO–cGMP–PKG pathway between nociceptive and antinociceptive effects?
  7. Are nNOS, eNOS, iNOS up-regulated in nervous tissues in response to neuropathic pain in this model?
  8. Is the target of antinociceptive effects of BBHC the same as that of Zerumbone?

This is a rather straightforward study by experienced groups that address how traditional medicine show the antinociceptive effect in vivo (Curcuminoids (54) Zerumbone (10, 25), Chalcones (52), Zingiber zerumbet (53)). But the manuscript is similar to that of previous report (10) using zerumbone compound. Thus, they should present the novelty findings of BBHC compared to that of zerumbone in term of the mechanism how BBHC act in the antinociceptive effect.

Author Response

Response to Reviewer 3 Comments

Point 1: The authors state that we had previously demonstrated that a newly synthesised compound named, 2-benzoyl-6-(3-bromo-4-hydroxybenzylidene)cyclohexen-1-ol (BBHC), from a novel family of diarylpentanoids possessed significant inhibitory activity on NO production in IFN-γ/LPS-induced RAW 264.7 macrophages [14] (lanes 70-73). But the authors do not show up and misread the contents of this paper.

Response 1: We apologise for the mistake for citing the wrong reference and correction has been made. The correct reference for lines 70-73 is Ref. no. 23 in the reference list.

Point 2: They focused on the L-arginine-nitric oxide-cGMP-ATP-sensitive K+ channel pathway. Thus, they should determine the effect of BBHC on NOSs enzyme activity in vitro.

Response 2: Thanks for your suggestion. However, the present study is only focused only on the L-arginine-nitric oxide-cGMP-ATP-sensitive K+ channel pathway in animal behavioural model. The in vitro study of BBHC’s involvement on NOS enzymatic activity is currently underway in our research group.

Point 3: Is the antinociceptive effect produced by BBHC reversed by naloxone, an opioid receptor antagonist?

Response 3: Yes, the antinociceptive effect produced by BBHC is reversed significantly by naloxone. The manuscript consisting this result is currently under review of another journal.

Point 4: They would be better off using 7-Nitroindazole asides from L-NOARG to test the involvement of neuronal NOS on the antinociceptive effect.

Response 4: Thanks for your suggestion. In our study, we started with L-NOARG to test the involvement of BBHC in NOS inhibition because L-NOARG is a non-selective NOS inhibitor. The more selective NOS inhibitor (e.g. 7-Nitroindazole) will be used in the future research of this compound.

Point 5: What is the difference mechanisms and potency between BBHC and Zerumbone (10)?

Response 5: For your information, the paper (ref no.10) published by our colleagues focused on neuropathic pain model while this present study focuses mainly on acute pain model.

In our research group, both BBHC and zerumbone are the candidates to study antinociceptive activities. BBHC is a novel synthetic diarylpentanoid analogue, produced via structural modification of curcumin whereas zerumbone is a naturally occurring cyclic sesquiterpene present in Zingiber zerumbet.

Both BBHC and zerumbone showed significant inhibition in animal models of acute pain based on our previous studies. However, BBHC was able to inhibit pain at lower doses (0.1, 0.3, 1.0 and 3.0 mg/kg) than zerumbone (0.1, 1.0, 5.0 and 10.0 mg/kg) in screening tests such as acetic acid-induced abdominal writhing test and hot plate test [The findings for the preliminary study of BBHC are currently under review for another journal].

Besides, BBHC and zerumbone also showed different mechanisms of action to inhibit acute nociception especially in which BBHC showed more receptor-specific involvement for noradrenergic, serotonergic, cholinergic and GABAergic systems. In addition, BBHC was tested for its acute toxicity based on OECD guidelines and it has been classified as Category 5 according to the Globally Harmonised System (GHS) for the classification of chemicals as relatively low acute toxicity hazard [these findings are currently under review as well].

Altogether, we are strongly encouraged that BBHC is a more potent antinociceptive candidate than zerumbone in acute nociceptive model based on its efficacy and safe usage.

Point 6:  How they interpretate the discrepancy of the NO–cGMP–PKG pathway between nociceptive and antinociceptive effects?

Response 6: The NO-cGMP-PKG pathway is involved in both pain induction and analgesia. In the nociceptive condition, the activation of NO-cGMP-PKG pathway will lead to elevated NO, cGMP and PKG production that then contribute to pain sensation. The attenuation of NO-cGMP-PKG pathway will lead to reduced production of NO, cGMP and PKG and develop antinociception.

Point 7: Are nNOS, eNOS, iNOS up-regulated in nervous tissues in response to neuropathic pain in this model?

Response 7: We do not study neuropathic pain in this model because the pain model that we used in the present study is an acute pain model. However, the common practice in our research group is that the extension of the study scope will be carried out after the antinociceptive properties and mechanisms of action of the antinociceptive compounds are confirmed via acute pain models. Therefore, the examination of BBHC in chronic pain model such as chronic constriction injury (neuropathic pain model) is in our future plan for this compound.  

Point 8: Is the target of antinociceptive effects of BBHC the same as that of Zerumbone?

Response 8: As we mentioned in the response for Point 5, BBHC and zerumbone showed different mechanisms of action in antinociception. BBHC showed different involvement for noradrenergic, serotonergic, cholinergic and GABAergic receptors as compared to zerumbone.

Reviewer 4 Report

The manuscript  entitled  „The involvement of L-arginine-nitric oxide-cGMP-ATP-sensitive K+ channel pathway in antinociception of BBHC, a novel diarylpentanoid analogue ” written by H.M. Ong et al. provides  evidence that  the systemic administration of BBHC is able to establish significant antinociceptive effect in mice model of chemically-induced pain. The effect of BBHC could be mediated by the involvement of L-arginine-nitric oxide-cGMP-ATP-sensitive K+ channel pathway, without any potential sedative or muscle relaxant concerns.

The design of the study is clear and methods are well-chosen to answer research questions.

The study provides some interesting observations:

  • BBHC generated significant inhibition to nitric oxide production which indicates an indirect deactivation of NOS.
  • BBHC prevented the catalytic action of soluble guanylyl cyclase and interrupted the cGMP pathway.
  • BBHC was also found to be a potential ATP-sensitive K+ channel opener that subsequently leads to reduced membrane excitability.

Minor commnet:

  • Figure 4: Error bars in two columns are missing (each column represents the mean ± S.E.M. of 6 mice).

In my opinion, this is well designed and well written paper. Authors provide reliable scientific evidence supporting their working hypothesis.

I recommend this manuscript for publication in Molecules.

Author Response

Response to Reviewer 4 Comments

Point 1: Figure 4: Error bars in two columns are missing (each column represents the mean ± S.E.M. of 6 mice).

Response 1: Thanks for your comment. There are no error bars in the columns for vehicle and BBHC because all the animals (n=6) pre-treated with vehicle and BBHC managed to maintain their motor performance on the rota-rod machine for 120s.

Round 2

Reviewer 2 Report

Thank you for making the suggested corrections to your manuscript. 

Reviewer 3 Report

In this revised manuscript, the responses to the major points raised previously were adequate and now this is acceptable for publication in Molecules.